# Antibody to Endogenous Cardiotonic Steroid Reverses Vascular Fibrosis and Restores Vasorelaxation in Chronic Kidney Disease

**DOI:** 10.3390/ijms25168896

**Published:** 2024-08-15

**Authors:** Natalia I. Agalakova, Elena V. Mikhailova, Ivan A. Ershov, Olga V. Nadei, Arseny A. Pyankov, Michael M. Galagoudza, C. David Adair, Irina V. Romanova, Alexei Y. Bagrov

**Affiliations:** 1Sechenov Institute of Evolutionary Physiology and Biochemistry, 194223 St. Petersburg, Russia; nagalak@mail.ru (N.I.A.); elenamikhailova87@gmail.com (E.V.M.); olganadej@gmail.com (O.V.N.); aa.pyankov@yandex.ru (A.A.P.); romanova.irina@gmail.com (I.V.R.); 2Department of Obstetrics and Gynecology, St. Petersburg State Pediatric Medical University, 194100 St. Petersburg, Russia; ershov_ia@inbox.ru; 3Almazov National Medical Research Centre, 197341 St. Petersburg, Russia; galagoudza@mail.ru; 4Department of Obstetrics and Gynecology, University of Tennessee College of Medicine, Chattanooga, TN 37403, USA; 5Padakonn Pharma, Puskini 9, 20309 Narva, Estonia

**Keywords:** fibrosis, Fli1, marinobufagenin, Na/K-ATPase, chronic kidney disease, vasorelaxation

## Abstract

Marinobufagenin (MBG) is implicated in chronic kidney disease, where it removes Fli1-induced inhibition of the collagen-1. We hypothesized that (i) in nephrectomized rats, aortic fibrosis develops due to elevated plasma MBG and inhibited Fli1, and (ii) that the antibody to MBG reduces collagen-1 and improves vasodilatation. A partial nephrectomy was performed in male Sprague-Dawley rats. Sham-operated animals comprised the control group. At 5 weeks following nephrectomy, rats were administered the vehicle (*n* = 8), or the anti-MBG antibody (*n* = 8). Isolated aortic rings were tested for their responsiveness to sodium nitroprusside following endothelin-1-induced constriction. In nephrectomized rats, there was an increase in the intensity of collagen staining in the aortic wall vs. the controls. In antibody-treated rats, the structure of bundles of collagen fibers had ordered organization. Western blots of the aorta had lower levels of Fli1 (arbitrary units, 1 ± 0.05 vs. 0.2 ± 0.01; *p* < 0.001) and greater collagen-1 (arbitrary units, 1 ± 0.01 vs. 9 ± 0.4; *p* < 0.001) vs. the control group. Administration of the MBG antibody to rats reversed the effect of the nephrectomy on Fli1 and collagen-1 proteins. Aortic rings pretreated with endothelin-1 exhibited 50% relaxation following the addition of sodium nitroprusside (EC_50_ = 0.28 μmol/L). The responsiveness of the aortic rings obtained from nephrectomized rats was markedly reduced (EC_50_ = 3.5 mol/L) compared to the control rings. Treatment of rats with the antibody restored vasorelaxation. Thus, the anti-MBG antibody counteracts the Fli1-collagen-1 system and reduces aortic fibrosis.

## 1. Introduction

Endogenous cardiotonic steroids (CTS) inhibit Na/K-ATPase and regulate the monovalent ions balance and cell homeostasis [1,2]. By binding to the Na/K-ATPase, CTS can affect cell growth and differentiation, apoptosis, and proliferation [1,2]. Marinobufagenin (MBG) is a CTS implicated in the pathogenesis of several pathological states, including preeclampsia [3] and chronic kidney disease (CKD) [4,5,6,7]. A novel effect of CTS is their ability to induce intracellular signaling, leading to a loss of elasticity and vascular fibrosis [8,9]. 

One of the mechanisms of the pro-fibrotic effect of MBG in CKD is the inhibition of the activity of Fli1, a nuclear transcription factor and a negative regulator of collagen-1 synthesis [10]. Fli1 competes with another transcription factor, ETS-1, to maintain a balance between stimulation and repression of *the collagen-1* gene [11,12]. The Na/K-ATPase/Src/EGFR complex emerges as a signal cascade, which activates phospholipase C, resulting in the phosphorylation of PKCδ and its translocation to the nucleus [10,13]. In the nucleus, PKCδ phosphorylates Fli1, which withdraws the Fli1-induced inhibition of the collagen-1 promoter and increases procollagen expression and collagen production [10,11,12]. Previously, we demonstrated that the levels of MBG are elevated in CKD and that MBG is involved in the genesis of cardiac fibrosis, which complicates CKD [4,8,9]. Aortic stiffness and interstitial myocardial fibrosis are interrelated, and this association is accelerated in CKD; therefore, early changes occurring in the vasculature are especially important to understand how the pathogenesis of CKD is coming forth [6,7,14,15]. 

The goal of the present experiment was to comprehend the ability of the MBG/Fli1-dependent mechanism to induce vascular fibrosis in CKD, and therefore to extend our previous hypothesis for preeclampsia to the pathogenesis of chronic renal failure. Preeclampsia shares many risk factors that contribute to renal fibrosis [16]. Here, we show that in partially nephrectomized (PNx) rats, aortic fibrosis develops due to an elevated plasma MBG level, how it is unfolding via a Fli1-sensitive mechanism, and how the fibrotic aortae exhibit impaired vasorelaxant responses, which are reversed by monoclonal antibodies raised against MBG.

## 2. Results 

In Sprague-Dawley rats, PNx led to hypertension and a marked increase in plasma creatinine compared to the control values (Table 1). The administration of aMBG antibodies to PNx animals decreased the arterial blood pressure to almost normal levels (Table 1). At the same time, no significant difference was observed between the blood pressure of Sham animals treated with aMBG antibodies (98.5 ± 2 mmHg, n = 8) and Sham rats (*p* > 0.05, two-tailed Student *t*-test). 

Moreover, the development of kidney failure in PNx animals was accompanied by a two-fold rise of the MBG content in blood plasma in comparison to sham-operated rats (Figure 1F). 

As presented in Figure 1A, the aortic wall of PNx rats was thicker, its structure was loose, and the organization of elastic lamellae (Weigert method) was disrupted compared to that of Sham animals. In PNx rats treated with the aMBG antibody, the intensity of elastin staining was comparable to that of Sham animals, and the aortae looked normal with a parallel orientation of the elastic material. The thickness of the aortic wall measured in the Weigert-stained preparations increased in PNx rats, but was similar to that of sham-treated animals (Figure 1C). The area occupied by elastin in the aortic wall was as follows: Sham—71.7%, PNx—41.3%, PNx + aMBG—59.6%, which indicates a decrease in the level of elastin in PNx rats compared to Sham and an increase in PNx + aMBG rats compared to the PNx group (Figure 1A,E). The results of Masson’s trichrome staining for collagen (Figure 1A, blue color) show that the adventitia of the aortic wall of PNx rats was thicker and denser, and closely adjacent to the tunica media compared to sham-operated rats. In aMBG-treated rats, the structure of the aortic adventitia was looser, and the bundles of collagen fibers had a predominantly ordered organization (Figure 1A). Statistical analysis demonstrates that the optical density of collagen-1 in the aortic wall of PNx rats is significantly higher than in the Sham group by (55%). After administration of anti-MBG, collagen-1 levels in the aortic wall were reduced by 38% compared with PNx, but remained 17% higher than in the Sham group (Figure 1B,D).

Compared to sham-operated animals within 5 weeks of PNx, the tissues of thoracic aortae had lower levels of Fli1 (Figure 2A), but a higher content of pro-collagen-1 (Figure 2B). The expression of collagen-1 in aortic tissues was increased in the thoracic aorta at the translational level (Figure 2C) and in the abdominal aorta at the transcriptional level (Figure 2D). However, no changes in the TGFβ and SMAD2 levels were observed (Figure 2E,F). Administration of aMBG mAb to PNx rats completely reversed the effect of the nephrectomy on Fli1 and levels of collagen-1 mRNA, and it significantly reduced levels of pro-collagen and collagen-1 proteins. At the same time, there was no considerable difference in Fli1 and collagen-1 immunoreactivity between Sham animals and those injected with aMBG mAb, in contrast to PNx rats (Appendix A).

Figure 3 presents the effects of PNx on the content of pro-fibrotic markers in the left ventricular myocardium. Fli1 was inhibited by 25% in the myocardium of PNx rats, and its levels returned to control values following treatment with aMBG mAb (Figure 3A). The levels of procollagen and collagen-1 increased significantly following induction of PNx and decreased with aMBG mAb (Figure 3B,C).

As shown in Figure 4, the aortic rings pretreated with 10 nmol/L endothelin-1 exhibited 50% relaxation following the addition of sodium nitroprusside at a concentration of 0.28 μmol/L (EC_50_ = 0.28 ± 0.18 μmol/L). The responsiveness of the aortic rings obtained from rats with PNx was markedly reduced (EC_50_ = 3.5 ± 1.4 μmol/L; *p* < 0.01) compared to the control rings. Treatment of PNx aortic rings with aMBG mAb restored their ability to relax following the addition of nitroprusside (EC_50_ = 0.98 ± 0.49 μmol/L; *p* < 0.05).

## 3. Discussion

The present study demonstrates that the MBG-Na/K-ATPase-Fli1-collagen-1 system is implicated in the pathogenesis of CKD, and indicates that monoclonal antibody to MBG is capable of counteracting the pressor action of MBG. Antagonism of this deleterious effect of MBG by the monoclonal aMBG antibody may lead to the prevention of vascular fibrosis in CKD, a hallmark of renal failure. The goals of our present study were (i) to show that rats following partial PNx are developing fibrosis of the aorta due to the Fli1-sensitive mechanism, (ii) that vascular fibrosis in PNx rats is sensitive to treatment with anti-MBG monoclonal antibodies, and (iii) that isolated rings of the thoracic aortae of PNx rats would exhibit impaired responses to the relaxant effect of sodium nitroprusside, and this effect will be reversed by the antibody to MBG. 

In the pathologic chain of events, an inhibition of Src kinase and phospholipase C by MBG leads to the phosphorylation of Fli1 by PKCδ, followed by procollagen synthesis [10,12]. Fli1 is a member of the Ets oncogene family of proteins, and it normally competes with ETS-1 to balance between stimulating and repressing the promoter of the collagen gene [11]. This transcription factor has been shown to play a role in vascular fibrosis [17,18], and a direct inhibitory effect of Fli1 on collagen-1 synthesis has been demonstrated in human umbilical arteries and placentae [19], in rat cardiac fibroblasts [8,10], and pig LLC-PK1 renal cells [13]. In the present study, we observed that at five weeks after PNx, the Fli1 signaling occurring in the aorta was more robust than the changes taking place in the ventricular myocardium. While the levels of procollagen and collagen-1 in the left ventricles changed by approximately 25% (Figure 3), procollagen in the aorta increased four-fold and collagen-1 six-fold (Figure 2). These observations closely resemble previous findings of associations between elevated plasma MBG and Fli1-induced fibrosis in mice [20] and rats [4,9] with renal failure and in patients with preeclampsia [19,21]. Moreover, these observations confirm the data from previous studies showing that Fli1 controls both the induction of dermal fibrosis in systemic sclerosis [18,22,23] and the formation of the tumor matrix microenvironment [24], and regulates *Fli1* gene expression in skeletal muscle fibroblasts [25].

Heightened dietary salt intake in several experimental models, including aged rats, normotensive rats, and diabetic rats, is associated with a scenario in which MBG and Fli1 became activated [26,27]. The results obtained by Piecha et al. [28] demonstrated that high maternal salt intake during pregnancy had a long-lasting effect on central arterial remodeling, and elevated MBG concentrations were found in animals with fibrosis of the walls of different arteries. Because two signaling pathways are known to modulate the profibrotic effects of MBG, a question arises whether the profibrotic effect of MBG may be initiated via SMAD-dependent TGFβ1 signaling, which was not the case for preeclampsia and CKD, where a key mechanism is Fli1 dependent [8,10,29]. Remarkably, in the study by Piecha et al., the heart weight increased by 18%, fibrosis of the aorta increased by 60%, and arterial fibrosis was not correlated with TGF-β levels [28], which was in line with previous observations in the heart [10]. These findings illustrate the importance of early MBG-induced pro-fibrotic signaling in the aorta for overdue increased myocardial stiffness.

In addition to the anti-fibrotic effect, aMBG mAbs lowered blood pressure and exhibited several other beneficial outcomes (Table 1). Moreover, we found that in PNx rats, immunoneutralization of MBG was associated with the normalization of water intake and increased urinary volume. As a result of these effects, the rats exhibited a three-fold increase in natriuresis (Table 1). Finally, aMBG mAb reduced plasma creatinine concentration by more than 30%. The latter observation is in accord with earlier data demonstrating that serum creatinine rose in PNx rats in which the injection of 3E9 aMBG mAbs resulted in reduction of left ventricular fibrosis [9]. PNx is not the only condition when we observe vascular fibrosis, the other scenario is experimental preeclampsia [29,30]. When pregnant rats are placed on a high-salt intake, they develop the symptoms of preeclampsia, such as high blood pressure and elevated MBG content, proteinuria, and fetal low weight and number [30]. After a single administration of 3E9 aMBG, blood pressure decreased for 48 h [29]. In patients with preeclampsia, MBG was elevated, vascular Fli1 was depressed, their umbilical arteries were stiff and fibrotic, and they failed to relax following enothelin-1-induced constriction [29]. Moreover, when umbilical arteries from normotensive pregnant subjects were treated for 24 h with nanomolar MBG, Fli1 in umbilical arteries was inhibited, levels of collagen-1 were increased, and following endothelin-1 contracture, they did not relax with sodium nitroprusside [29,31]. Furthermore, when in healthy umbilical arteries, the levels of Fli1 were reduced by siRNA, mimicking PE, and the synthesis of collagen-1 dramatically increased [31]. In accordance with these findings, we observed that MBG is produced by cultured human placental JEG 3 cells [32].

In brief, our study shows that in PNx rats, the fibrosis of the aorta is a result of inhibition of Fli1 in that vascular tissue. In rats with renal failure, aortic fibrosis is sensitive to treatment with the anti-MBG monoclonal antibody, and isolated rings from thoracic aortae of CKD rats exhibit impaired responses to the relaxant effect of sodium nitroprusside, and these effects were reversed by the antibody to MBG. Importantly, in rats with CKD, the fibrosis of the aorta precedes myocardial fibrosis, and the same observation was made previously in CKD patients [14]. Stiffening of the aortic wall and improper matching between aortic diameter and flow are frequently associated with changes in pulsatile hemodynamics, including an increase in arterial pressure wave amplitude, which increases pulse pressure [33]. The antagonism of the profibrotic effect of MBG by monoclonal aMBG mAb may lead to the prevention of vascular fibrosis in CKD, a distinctive feature of renal failure in rats [10] and patients with end-stage renal disease [34]. 

## 4. Materials and Methods

**General:** Animal experiments were conducted in accordance with the National Institutes of Health (NIH) Guide for the Care and Use of Laboratory Animals (8th Edition, 2011). The animal study design was approved by the Institutional Review Board of Almazov National Cardiology Centre (DT-IAC001-v2.0-Jan 2019) and by the Bioethics Committee of the Institute of Evolutionary Physiology and Biochemistry of the Russian Academy of Sciences (protocol # 12/2020 of 24 December 2020), St. Petersburg, Russia. Two-stage PNx was performed in male Sprague-Dawley rats (250–300 g) as reported previously [8,19]. First, a subtotal nephrectomy of the left kidney was performed, and 2 weeks later, the right kidney was totally removed. Eight rats comprised the control (Sham operated) group. At 4 weeks following the second step of PNx, PNx rats were intraperitoneally administered the vehicle (isotype IgG1, *n* = 8), or monoclonal anti-MBG mAb (*n* = 8). The injection of IgG1 and aMBG mAbs was repeated after 7 days. The dose of anti-MBG 4G4 mAb (100 mg/kg) was the same as that previously reported to reverse the EC_75_ to the inhibition of the Na/K-ATPase by MBG [30]. In addition, we gave a one-time intraperitoneal injection of 100 mg/kg anti-MBG 4G4 mAb (100 mg/kg) to four Sham rats. Number of observations means the number of animals/tissues that were analyzed by a given method. Each analyzed group contained 8 rats, however, the tissues of 5 animals from each were used for histochemistry, Western blotting, and other experiments. At 5 weeks after the second stage of PNx, 24-h urine samples were collected, and blood pressure was determined using the tail-cuff method. Then, the rats were anaesthetized by 50 mg/kg Isoflurane (Abbot, UK) and sacrificed by exsanguination. Blood samples were obtained and plasma creatinine, Na, and K concentrations were determined (Hospitex Diagnostics, **Florence**, Italy, Accent-200, PZ Cormay, Guangzhou, China). Plasma concentrations of MBG were measured as described previously [8].

**Preparation of the aorta sections.** The pieces of the abdominal aorta were fixed for 72 h (4 °C) in 4% p-formaldehyde solution (Sigma, Burlington, MA, USA), washed with 0.9% Na-phosphate buffer (PBS) and immersed in PBS containing 30% sucrose at 4 °C. Then, the pieces were frozen on dry ice using the Tissue-Tek^®^ medium (Sakura Finetek Europe, Alphen aan den Rijn, The Netherlands) and stored at −80 °C. Cross-sections of 10 µm from different levels of the aorta were prepared using a Leica CM-1510 cryostat (Leica Microsystems, Wetzlar, Germany). The sections from Sham, PNx, and PNx + aMBG groups were mounted on the glasses (SuperFrost/plus, Menzel, Germany), dried overnight at room temperature, and used for Weigert staining for identifying elastic structures, or Masson’s trichrome staining for identification of collagens, or stored at −20 °C before immunohistochemical reactions.

**Histological analysis.** After Weigert (Weigert long method for elastic fibers kit, Bio-Optica, Milan, Italy) and Masson’s (Masson’s trichrome stain kit, BioVitrum, St-Petersburg, Russia) staining, the slides with aortic segments were washed in distilled water, and after standard histological processing, they were cover slipped in the Bio-Mount medium (Bio-Optica, Italy) and dried.

**Immunohistochemistry.** The slides with aortic sections were dried overnight at room temperature, washed with PBS, treated with 1% hydrogen peroxide in PBS for 30 min to block endogenous peroxidase activity, washed with PBS for 15 min, and then treated with PBS containing 0.1% Triton X-100 (PBST) for 30 min. Subsequently, the sections were incubated for 1 h in a blocking solution (mix of 3% goat and 2% bovine serum in PBST). Incubation with the primary monoclonal mouse anti-Collagen-1A Antibody (sc-59772, Lot #12718, Santa Cruz Biotechnology, Seattle, WA, USA) was carried out in a 2% blocking solution at a dilution of 1:200 48 h (4 °С). After washing with PBST (40 min), the sections were incubated for 1 h in PBST with the rabbit anti-mouse peroxidase-conjugated IgG (A9044, Lot# 010M4797, Sigma-Aldrich, St. Louis, MO, USA) at a dilution of 1:400 for 1 h. After washing in PBS, the sections were treated with 0.05% diaminobenzidine (Sigma-Aldrich, St. Louis, MO, USA) and 0.03% hydrogen peroxide in PBS. The reaction was stopped in distilled water, and after washing, the sections were stained with hematoxylin and placed under a glass cover using glycerol. The specificity of the immunohistochemical reaction was checked using a negative control (the samples without the primary or secondary antibodies). 

**Microscopy.** The micrographs of different sections of the aorta were obtained using a transmitted light Carl Zeiss Imager A1 (Axio Vision 4.7.2) microscope, at ×20 magnification(Carl Zeiss, Oberkochen, Germany) using the same optic characteristics for different groups of animals. In the Weigert stained sections, the thickness of the media was determined in µm in different opposing areas of the aorta section in each microphotograph, and the area of elastin in the media and the total media area was determined using the Image J NIH Analysis software (https://imagej.net, 25 July 2024)(National Institutes of Health, Bethesda, MD, USA). The elastin area is expressed as a percentage relative to the total media area. The optical density of collagen-1-immunopositivity was quantified using the Image J NIH Analysis software (https://imagej.net, 25 July 2024) (National Institutes of Health, Bethesda, MD, USA) and was estimated and expressed in arbitrary units.

**Western blot** analysis was performed on proteins from the thoracic aorta and left ventricular myocardium homogenates as reported previously [29]. The fragments of aortas and isolated heart left ventricles were immediately frozen on dry ice and stored at −80 °C until processing. The tissues were minced and homogenized in an ice-cold RIPA lysis buffer supplemented with a protease and phosphatases inhibitor cocktail (sc-24948, Santa Cruz Biotechnology Inc., USA). The homogenates were centrifuged for 15 min at 11,000× *g* and 4 °C, and the resulting supernatants were collected and frozen at –80 °C until use. The content of total protein in the samples was determined by the Lowry method. The proteins in the samples were solubilized by boiling with the Laemmli sample buffer and separated by SDS–PAGE in 10% polyacrylamide gels with each well loaded with an equal amount of total protein (50 µg/well). Then, the resolved proteins were transferred to nitrocellulose membranes, and the non-specific binding was blocked with 5% non-fat dry milk diluted in TTBS (0.1% Tween-20 in Tris-buffered saline) for 1 h at room temperature. Then, the membranes were incubated overnight, with the primary rabbit recombinant monoclonal Fli1 antibodies [EPR4645] (1:500, Catalogue number ab124791, Lot GR80733-12, Abcam, Inc., Waltham, MA, USA), mouse monoclonal antibodies to pro-COL1A2 (D-6) (1:500, sc-166572, Lot #E0718, Santa Cruz Biotechnology, Dallas, TX, USA), mouse monoclonal antibodies to COL1A (1:500, sc-59772, Lot #12718, Santa Cruz Biotechnology), rabbit polyclonal TGF-β (1:500, #3711, Lot 7, Cell Signaling Technology Inc., Danvers, MA, USA), and rabbit polyclonal SMAD2/3 (D7G7) (1:500, #8685, Lot 6, Cell Signaling Technology). All the antibodies used are recommended by the manufacturers as highly specific and reacting with the proteins of human, rat, and mouse origin. After washing, the membranes were probed for 2 h with the secondary Amersham ECL anti mouse HRP-linked whole antibody from sheep (1:2000, NA931) or anti-rabbit HRP linked whole antibody from donkey (1:1000, NA934, Lot 16831195, Cytiva, Wilmington, DE, USA). Immunostained proteins were visualized by exposure with the chemiluminescent Amersham ECL detection system (RPN3004, Cytiva). To control protein loading, the membranes were striped by two rounds of 30-min incubation with an acidic solution (25 mM glycine, 1% SDS, pH 2.0) followed by 10-min washing with TBS, and re-probed with mouse antibodies against GAPDH (1:2000, sc-32233, Santa Cruz Biotechnology), followed by secondary HRP-conjugated antibodies. The relative protein expression was evaluated by densitometric analysis using the ImageJ program (NIH, Bethesda, USA). The optical densities of target proteins were normalized to that of reference GAPDH proteins. 

**Real-time PCR of collagen-1 gene expression:** *RNA extraction, reverse transcription, and real-time quantitative PCR analysis were performed as described [35].* Gene expression levels were analyzed using a quantitative real-time PCR. Total RNA was isolated from abdominal aorta samples using the ExtractRNA reagent, and RNA concentration and purity (A260/A280 ≥ 1.6) were determined using an Implen C40 NanoPhotometer (Olis, Athens, GA, USA). Reverse transcription (RT) synthesis was performed with 1 μg total RNA and the RT MMLV RT kit. Amplification was performed in a mixture (25 μL) containing 10 ng of the RT product, by 0.4 μM of forward (F) and reverse (R) primers, and a qPCRmix-HS SYBR + LowROX reaction mixture in 96-well PCR plates (in triplicate) on a Real-Time System CFX96 C1000 Touch Thermal Cycler instrument (BioRad Laboratories, Singapore). The PCR product purity was tested by electrophoresis in a 30% agarose gel with ethidium bromide and assessed using a gel documentation system (Chemidoc, BioRad, UK). The delta-delta Ct method was expressed as fold expression relative to expression in the Sham group. The primers to the *Collage-1* gene (NM_053304.1, F—TGG CAA CCT CAA GAA GTC CC, R—ACA AGC GTG CTG TAG GTG AA) and the reference *18s rRNA* gene (NR_046237.1, F—GGACACGGACAGGATTGACA, R—ACCCACGGAATCGAGAAAGA) were used. PCR results demonstrate the same number of cycles during amplification of the 18S rRNA product in the three studied groups; the Ct level for *18S rRNA* in the studied groups did not differ, which indicates that the injections of antibodies to MBG does not affect the control gene expression.

**Isolated rat aorta contractile studies**: Explants of rat thoracic aortae were washed 3 times in fresh incubation media [36]. Rings of rat aortae (2 mm) were suspended at a resting tension of 1.0 g in a 15-mL organ bath (Ugo Basile, Italy) and superfused at 37 °C with a solution containing in mmol/L: NaCl 130, KCl 4.0, CaCl_2_ 1.8, MgCl_2_ 1.0, NaH_2_PO_4_ 0.4, NaHCO_3_ 19, and glucose 5.4, then gassed with a mixture of 95% O_2_ and 5% CO_2_ (pH 7.45). Isometric contractions were recorded as reported previously. The aortic rings were constricted twice with 80 mmol/L of KCl. Next, we studied the vasorelaxation of aortic rings by sodium nitroprusside (1–10 μmol/L) following constriction of vascular rings with 100 nmol/L endothelin-1, as described previously [37]. The force of contractions was expressed as the percent of vasoconstrictor response to 80 mmol/L KCl. The percent of relaxation was calculated relative to the plateau of contractile force that was achieved in response to 100 nmol/L endothelin-1. The maximal force of contraction of the aortic rings to 100 nmol/L endothelin-1 was 1.7 ± 0.09 g in the Sham group, 1.6 ± 0.16 g in the PNx group, and 1.8 ± 0.17 g in the PNx + aMBG group (n = 5).

**Statistical analyses**: Results are reported as the mean ± standard error of the mean (SEM). The significance of differences among the measured variables was assessed by a two-tailed *t*-test, one-way analysis of variance (ANOVA) followed by the Tuckey’s multiple comparison test, or by two-way ANOVA followed by Holm-Sidak’s multiple comparisons test (GraphPad Prism software 6; GraphPad Inc., San Diego, CA, USA). A two-sided *P* value of <0.05 was considered significant. 

## Figures and Tables

**Figure 1 ijms-25-08896-f001:**
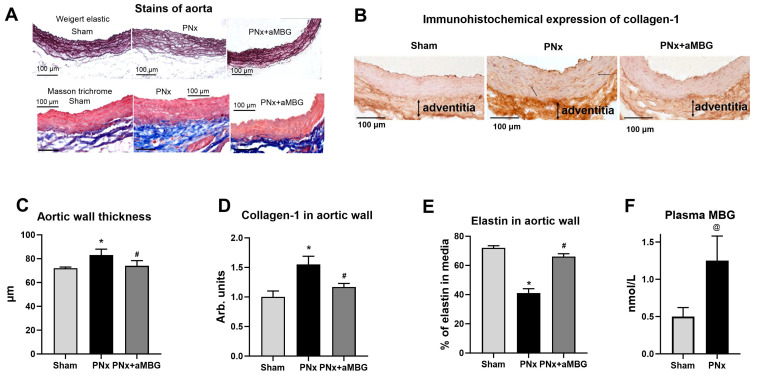
Effect of PNx and treatment of PNx rats with anti-MBG mAb on elastin and collagen content in abdominal aorta at five weeks after surgery: (**A**) representative images of aortae sections stained according to Weighert and Masson’s methods; (**B**) typical images of aortae sections after immunohistochemical staining for collagen-1; (**C**) thickness of the aortic wall measured in preparations stained according to Weigert; (**D**) levels of collagen-1 in aortic wall calculated by immunohistochemical method; (**E**) area occupied by elastin in aortic wall evaluated after Weigert staining; (**F**) plasma MBG content at 5 weeks post-surgery. Statistical analysis: one-way ANOVA followed by Tuckey’s multiple comparison test: *—*p* < 0.001 vs. Sham, #—*p* < 0.001 vs. PNx. ((**C**–**E**) n = 5), two-tailed Student *t*-test, @ *p* < 0.01 ((**F**) n = 8).

**Figure 2 ijms-25-08896-f002:**
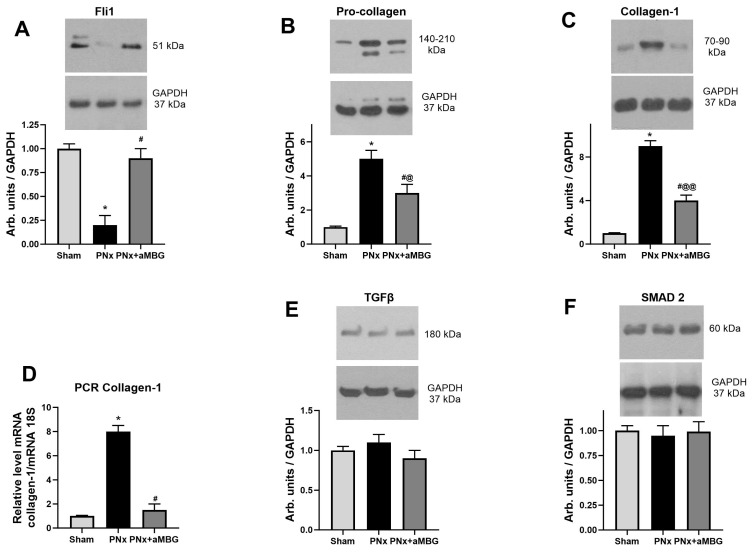
The levels of Fli1 (**A**), pro-collagen (**B**), collagen-1 protein (**C**), TGFβ (**E**), and SMAD2 (**F**) in thoracic aortae (n = 5 for (**A**–**C**,**E**,**F**)) and content of collagen-1 mRNA (**D**) in abdominal aortae (n = 8) of sham rats, untreated PNx animals, and PNx rats treated with anti-MBG mAb. Each bar represents mean ± SEM from five observations. One-way ANOVA followed by Tuckey’s multiple comparison test: *—*p* < 0.001 vs. Sham; #—*p* < 0.001 vs. PNx; @—*p* < 0.05, @@—*p* < 0.001 vs. Sham.

**Figure 3 ijms-25-08896-f003:**
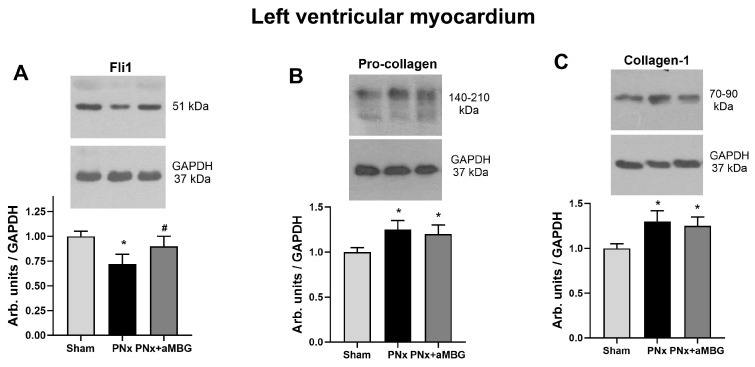
The effect of anti-MBG mAb on the levels of Fli1 (**A**), pro-collagen (**B**), and collagen-1 (**C**) in the left ventricular myocardium of sham-treated and PNx rats. Each bar represents mean ± SEM from five observations. One-way analysis of variance followed by Tuckey’s test: *—*p* < 0.01 vs. Sham, #—*p* < 0.01 vs. PNx.

**Figure 4 ijms-25-08896-f004:**
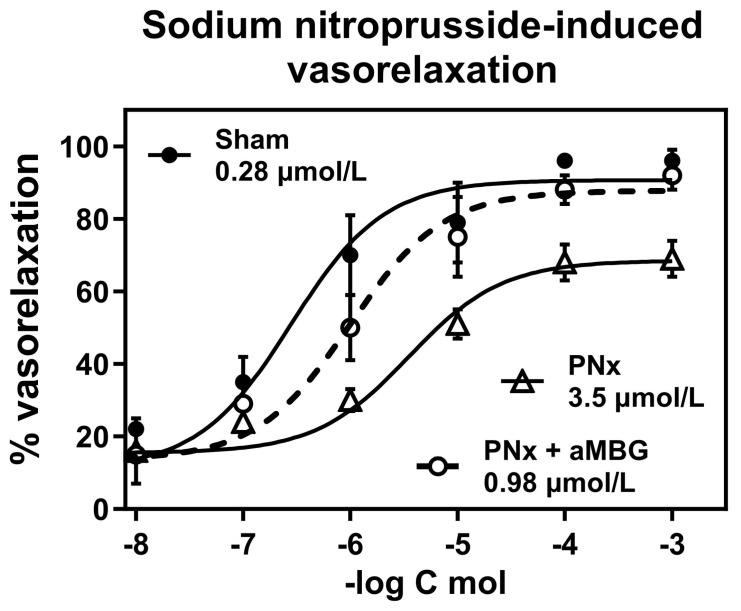
Responsiveness of thoracic aortic rings to sodium nitroprusside-induced vasorelaxation following contractions induced by 100 nmol/L endothelin-1. Concentration–response curves were determined five times and averaged to give the means and SEM for IC_50_. Two-way ANOVA: Interaction *p* = 0.0103, Row factor—*p* < 0.0001, Column factor—*p* < 0.0001. Multiple comparisons test for two-way ANOVA: Sham vs. PNx—*p* < 0.0001; and PNx vs. PNx + aMBG—*p* < 0.001; Sham vs. PNx-aMBG—*p* > 0.05.

**Table 1 ijms-25-08896-t001:** Physiological parameters of rats after PNx and PNx treated with antiMBG mAbs.

	Sham (n = 8)	PNx (n = 8)	PNx + anti-MBG (n = 8)
Blood pressure (mm Hg)	101 ± 1	149 ± 2 ***	114 ± 2 ***^,@@@^
Body weight (g)	309 ± 9	298 ± 10	278 ± 10
LV weight/ body weight (g/kg)	0.121 ± 0.008	0.138 ± 0.008	0.144 ± 0.0016
Right kidney weight (g)	1.09 ± 0.06	1.12 ± 0.06	1.1 ± 0.05
Water intake (mL/24 h)	31.7 ± 2.0	45.5 ± 2.3 ***	37.0 ± 1.2 ^@^
Urine volume (mL/24 h)	13.0 ± 1.3	29.4 ± 1.9 ***	16.4 ± 1.2 ^@@^
Plasma creatinine (μmol/L)	22.8 ± 3.2	119 ± 5.5 ***	80.8 ± 5.5 ***^,@@@^
Plasma Na (mmol/L)	146 ± 1	143 ± 1	144 ± 4
Plasma K (mmol/L)	6.6 ± 0.3	7.9 ± 0.29 **	8.1 ± 0.3 **
Urinary Na excretion (mmol/L)	25.5 ± 3.7	8.9 ± 1.1 *	23.4 ± 4.5 ^@^

Systolic blood pressure was measured in conscious animals with a tail cuff at 5 weeks post-surgery; LV, left ventricular. One-way ANOVA followed by Tuckey’s multiple comparisons test: *—*p* < 0.01; **—*p* < 0.001 and ***—*p* < 0.0001 vs. Sham-operated rats, ^@^—*p* < 0.01; ^@@^—*p* < 0.001 and ^@@@^—*p* < 0.0001 vs. PNx rats.

## Data Availability

The data that support the findings of this study are available from the corresponding author upon reasonable request.

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
