# Peer review of "Antibody to Endogenous Cardiotonic Steroid Reverses Vascular Fibrosis and Restores Vasorelaxation in Chronic Kidney Disease"

_ijms, 2024, doi:10.3390/ijms25168896_

Round 1
Reviewer 1 Report
Comments and Suggestions for Authors
The manuscript „Antibody to endogenous cardiotonic steroid reverses vascular fibrosis and restores vasorelaxation in chronic kidney disease“ by Natalia I. Agalakova, Elena V. Mikhailova, Ivan A. Ershov, Olga V. Nadei, Arseny A. Pyankov, Michael M. Galagoudza, C. David Adair, Irina V. Romanova and Alexei Y. Bagrov describes new findings showing that in partially nephrectomized (PNx) rats aortic fibrosis is observed, associated with increased MBG and possibly mediated by the Fli1-pathway. In addition, vasorelaxation to SNP is impaired. Many of these alterations are reversed by an antibody against MBG.
This referee has the following comments:
- Introduction: The particular knowledge gap to be addressed is missing. Also, the specific hypothesis of the study, derived from the knowledge gap, is not provided (or the explanation that the present study is hypothesis-generating). Please add.
- Method, general: The version of the National Institutes of Health (NIH) Guide for the Care and Use of Laboratory Animals should be provided. The permission for the animal experiments by your local authorities should be provided here. The phrase “and these procedures with repeated in 7 days“ is unclear.
- Methods, Western blotting: Information on antibodies is partly incomplete, e.g. lot numbers; in particular, evidence for the specificity of the antibodies should be provided.
- Methods, contractile studies: There are a number of spelling errors in the description of the solution composition, please correct. The use of 1g resting tension should be justified (provide evidence that this is the tension providing optimal contraction conditions). Description of the test for successful removal of the endothelium is missing, please add.
- Methods, statistics: A statement of how you defined biological replicates is missing, please provide.
- Results, general: One important control group seems to be missing, sham with MBG antibody, to test the effect of the antibody itself. How can you be sure that it is the interaction of the antibody with MBG and not the antibody itself producing the effects in the treated group? Please provide data or a convincing argumentation.
- Results, table 1: The statement in the legend “Systolic blood pressure measured in conscious animals with a tail cuff at the indicated time points” is unclear as there are no time points indicated in the table. The symbol ### is not described. Please explain. As p<0.05 was given as the limit for statistical significance, the statements p<0.01, p<0.001 and p<0.0001 are statistically incorrect, please revise.
- Results, Fig. 1: For panel A the n-number is missing. The statement “Plasma levels of MBG in PNx were elevated 4-fold vs. that in sham-operated animals” is unclear as there the number for plasma MBG seems to be 0.5 and about 1.25. In panel B only examples are shown. Thus, effect size and reproducibility of these data is unclear. Please provide a quantification and statistical analysis for the parameters analyzed, wall thickness, structure and intensity of elastin membranes, collagen and structure of the adventitia.
- Results, Fig. 2: For the example blots, full length blots including the molecular weight marker lane should be provided. As GAPDH was used for normalization, please provide evidence that PNx and MBG treatment does not alter GAPDH expression. Was the PCR signal also normalized? For collagen and pro-collagen, was the treated group different from sham? The statement “Compared to sham-operated animals within 5 weeks” is unclear, wasn’t it 7 weeks? For the statement “Administration of aMBG mAbs to rats almost completely reversed the effect of PNx on Fli1, pro-collagen, and collagen-1 proteins.” p-values for the comparisons made should be provided (also for collagen PCR).
- Results, Fig. 3: For the example blots, full length blots including the molecular weight marker lane should be provided. The statement of the effects MBG treatment vs sham in panel A, B, and C should be justified statistically. Where is the evidence for the statement “The magnitude of these effects in the left ventricular myocardium was much less than in the aorta”?
- Results, Fig. 4: What was the reason to test only one vasodilator, SNP and not a vasoconstrictor? For an appropriate comparison of the effect of SNP between the groups, the endothelin-1-induced pre-constriction (in % of maximum constriction induced by 80mM KCl) should be provided. In addition these levels should be compared between groups. Were endothelin-1 concentration response relationships different between groups? Did you use 10nm (text) or 100nM (legend) endothelin-1? The 3 groups should be compared pairwise among each other. Which groups did you compare for the statement “Treatment of PNx aortic rings with aMBG mAb restored their ability to relax following the addition of nitroprusside”? Why didn’t you consider to use a repeated measures ANOVA, as far as I understand, SNP was given cumulatively? What do you mean with “Concentration-response curves were determined 5 times”, were this technical or biological replicates?
- Discussion: The statement “In the present study, we observed that at seven weeks of PNx the Fli1 signaling occurring in the aorta preceded changes taking place in the ventricular myocardium” is not justified as data from different time points showing the dynamics of the changes induced are not provided.
- Discussion: The statement “As a result of these effects, rats exhibited a four-fold increase in natriuresis (Table 1).” is unclear. The numbers don’t show a four-fold increase.
Author Response
We are most grateful to the Reviewer for the insightful critique. We apologize for oversights and we adjusted text as suggested. Text changes are highlighted.
· Introduction: The particular knowledge gap to be addressed is missing. Also, the specific hypothesis of the study, derived from the knowledge gap, is not provided (or the explanation that the present study is hypothesis-generating). Please add.
|
This sentence is added. Lines 71-72. |
· Method, general: The version of the National Institutes of Health (NIH) Guide for the Care and Use of Laboratory Animals should be provided. The permission for the animal experiments by your local authorities should be provided here.
|
The number of NIH Guide version and the number of approval protocol of our Institutes were added to the text. Lines 228-231.
|
The phrase “and these procedures with repeated in 7 days“ is unclear.
|
The phrase was changed to “The injection of IgG1 and a/MBG mAb was repeated in 7 days”. Lines 234-238.
|
· Methods, Western blotting: Information on antibodies is partly incomplete, e.g. lot numbers; in particular, evidence for the specificity of the antibodies should be provided.
|
The lot numbers of the antibodies used in the study was added to the text. As stated by the manufacturers, all antibodies are highly specific and react with proteins of human, rat and mouse origin. All antibodies recognized only one band at correct molecular weight. Lines 290-302.
|
· Methods, contractile studies: There are a number of spelling errors in the description of the solution composition, please correct. The use of 1g resting tension should be justified (provide evidence that this is the tension providing optimal contraction conditions). Description of the test for successful removal of the endothelium is missing, please add.
|
Two papers are quoted (35,36). Respectfully, we do not fully agree. Different authors use resting tension of 1 or 2 grams. We use 1 gram. Why? For example, in one of our papers (PMID: 10694190), when the tension was 1 gram, cicletanine (a vasorelaxant) was inhibiting PKC with IC50 of 40 umol/L, while elsewhere (PMID: 1700224) when the tension was 2 grams the PKC IC50 of the same drug was increased by many times and reached 900 uomol/L! Therefore, optimal contraction ability may lead to over stretching the vessel. Endothelium was removed via gently rubbing inside the lumen of the vessel ring using a fine pair of small forceps (Lines 326-327). |
· Methods, statistics: A statement of how you defined biological replicates is missing, please provide.
|
This information was added to the subsection “Statistical analysis”. Number of observations (or n) means the number of animals the tissues of which were analyzed by a given method. Each analyzed group contained 8 rats, however, the tissues of 5 animals from each were used for histochemistry, western blotting and other experiments. Lines 335-336, 339-340.
|
· Results, general: One important control group seems to be missing, sham with MBG antibody, to test the effect of the antibody itself. How can you be sure that it is the interaction of the antibody with MBG and not the antibody itself producing the effects in the treated group? Please provide data or a convincing argumentation.
|
Actually, there was Sham + aMBG Ab group in our study. However, administration of antibodies to Sham animals did not change the studied parameters. Thus, SPB of Sham rats did not differ statistically significantly after aMBG Ab administration (Fig. 1 Line 484-494, addendum). aMBG Ab did not change the level of Fli1 in Sham animals, in contrast to PNx rats (Methods, line 237-238). |
· Results, table 1: The statement in the legend “Systolic blood pressure measured in conscious animals with a tail cuff at the indicated time points” is unclear as there are no time points indicated in the table.
|
We apologize for this error. The legend was re-written.
|
The symbol ### is not described. Please explain. As p<0.05 was given as the limit for statistical significance, the statements p<0.01, p<0.001 and p<0.0001 are statistically incorrect, please revise.
|
The symbol ### was an error.
|
· Results, Fig. 1: For panel A the n-number is missing.
|
The number of analyzed animals was added to legend. Lines 106-111.
|
The statement “Plasma levels of MBG in PNx were elevated 4-fold vs. that in sham-operated animals” is unclear as there the number for plasma MBG seems to be 0.5 and about 1.25.
In panel B only examples are shown. Thus, effect size and reproducibility of these data is unclear. Please provide a quantification and statistical analysis for the parameters analyzed, wall thickness, structure and intensity of elastin membranes, collagen and structure of the adventitia.
|
It was corrected.
Actually, in the first we planned to use the Figure only as a representative image. We added missing data to Figure 3. |
· Results, Fig. 2: For the example blots, full length blots including the molecular weight marker lane should be provided. As GAPDH was used for normalization, please provide evidence that PNx and MBG treatment does not alter GAPDH expression.
|
We used the classical manual method of developing X-ray films with developer and fixer solutions. Since the standard molecular weight proteins do not bind to ECL solutions mixture, they do not appear on X-ray film. The correct location of the bands of the studied proteins relative to the bands of the marker proteins was confirmed by manual overlay of X-ray film to the membranes. So, we cannot provide the blots with molecular weight marker lane unless by manually drawn. The original full-length blots were provided at the time of submission as required by the IJMS. However, we think that presenting of full-length blots with large empty space is not reasonable since this can greatly overload the figures.
|
Was the PCR signal also normalized? For collagen and pro-collagen, was the treated group different from sham?
|
The relative level of Collagen-1 mRNA was determined relative to the level mRNA 18S - the control, the delta-delta Ct method (Methods are rewritten, lines 311-325) was used, so that the results were normalized. In the Figure 2D Arb.units (Y axis) was changed to Relative level mRNA Collagen-1/mRNA 18S (lines 122-125). |
The statement “Compared to sham-operated animals within 5 weeks” is unclear, wasn’t it 7 weeks?
|
We apologize for this uncertainty. Since nephrectomy was performed in two steps, 7 weeks means from the first stage – left subtotal PNx, while 5 weeks means from the final total right PNx. To avoid such discrepancy, we changed “7 weeks” to “5 weeks” throughout the text.
|
For the statement “Administration of aMBG mAbs to rats almost completely reversed the effect of PNx on Fli1, pro-collagen, and collagen-1 proteins.” p-values for the comparisons made should be provided (also for collagen PCR). |
We respectfully point on the values of statistical significance, Fig.2, footnotes: By 1-way analysis of variance followed by Tuckey test: * - P < 0.001 vs. Sham, # - P < 0.001 vs. PNx.
|
• Results, Fig. 3: For the example blots, full length blots including the molecular weight marker lane should be provided. |
We cannot provide full length blots including the molecular weight marker lane due to the same reasons as for Figure 2. |
The statement of the effects MBG treatment vs sham in panel A, B, and C should be justified statistically. Where is the evidence for the statement “The magnitude of these effects in the left ventricular myocardium was much less than in the aorta”? |
This sentence has been omitted. This issue is already addressed in the Discussion (line 328) |
• Results, Fig. 4: What was the reason to test only one vasodilator, SNP and not a vasoconstrictor? For an appropriate comparison of the effect of SNP between the groups, the endothelin-1-induced pre-constriction (in % of maximum constriction induced by 80mM KCl) should be provided. In addition these levels should be compared between groups. Were endothelin-1 concentration response relationships different between groups? Did you use 10nm (text) or 100nM (legend) endothelin-1? The 3 groups should be compared pairwise among each other. Which groups did you compare for the statement “Treatment of PNx aortic rings with aMBG mAb restored their ability to relax following the addition of nitroprusside”? Why didn’t you consider to use a repeated measures ANOVA, as far as I understand, SNP was given cumulatively? What do you mean with “Concentration-response curves were determined 5 times”, were this technical or biological replicates? |
The main reason was because using a vasodilation is the idea of the study, and of several previous publications (PMID: 35328757, PMID: 31415452, PMID: 26350300, PMID: 26136067, PMID: 21330936). We think that inability to relax is extremely important. Date on vasoconstrictor effect of endothelin-1 are presented in the Methods (lines 335-336). We used endothelin-1 at concentration 100 nM, sorry for this oversight. We used two-way ANOVA the numbers are in the footnotes.
“Concentration-response curves were determined 5 times” means that totally 5 times the triad (Sham, PNx, PNx+aMBG) has been run. The runs which were not full do not count. |
· Discussion: The statement “In the present study, we observed that at seven weeks of PNx the Fli1 signaling occurring in the aorta preceded changes taking place in the ventricular myocardium” is not justified as data from different time points showing the dynamics of the changes induced are not provided. |
It has been rephrased (line 178). |
· Discussion: The statement “As a result of these effects, rats exhibited a four-fold increase in natriuresis (Table 1).” is unclear. The numbers don’t show a four-fold increase.
|
8.9 vs. 23.4 means the increase for 2.6 times. Sorry.
|
Reviewer 2 Report
Comments and Suggestions for Authors
The authors have tried to establish a link btw the transcriptional factor FLi1 and collagen in the aorta and tried to link it to CKD. However, there are many major concerns regarding the manuscript. Please find the comments in the attached documents.

Author Response
We are most grateful to the Reviewer for the insightful critique. We apologize for oversights and we adjusted text as suggested. Text changes are highlighted.
- Quantification in Fig 1.
This has been done.
- mRNA normalization.
It has been normalized.
- The magnitude of the heart ….
This has been rephrased. This sentence is rewritten. Line 178-181.
- Fix the spelling.
It has been done. Thank you.
Round 2
Reviewer 1 Report
Comments and Suggestions for Authors
The manuscript „Antibody to endogenous cardiotonic steroid reverses vascular fibrosis and restores vasorelaxation in chronic kidney disease“ by Natalia I. Agalakova, Elena V. Mikhailova, Ivan A. Ershov, Olga V. Nadei, Arseny A. Pyankov, Michael M. Galagoudza, C. David Adair, Irina V. Romanova and Alexei Y. Bagrov has been revised according to the comments of this referee.
This referee still some comments:
- Introduction: Your answer is appreciated. However, a goal is not a specific, testable hypothesis. Or is your study hypothesis-generating? Please provide this information.
- Methods, contractile studies: The description of the test for successful removal of the endothelium is still missing, please add.
- Methods, statistics: The explanation regarding the definition of biological replicates is appreciated. However, it seems not to be added to the statistics section. Please add.
- Results, Fig. 2: You did not answer to my question: “As GAPDH was used for normalization, please provide evidence that PNx and MBG treatment does not alter GAPDH expression.” Please provide.
- Results, Fig. 2: Your explanation that the PCR signal was normalized to 18S is appreciated. Please provide information that the expression of 18S is not affected by your treatments.
- Results, Fig. 2: You did not answer to my previous question “For collagen and pro-collagen, was the treated group different from sham?”
- Results, Fig. 2: Your answer to my question “For the statement “Administration of aMBG mAbs to rats almost completely reversed the effect of PNx on Fli1, pro-collagen, and collagen-1 proteins.” p-values for the comparisons made should be provided (also for collagen PCR).” is unclear. For this statement the effect of PNx should be calculated (delta (PNx - sham) and compared with the effect of treatment. Otherwise, your statement is without justification.
- Results, Fig. 4: The statistical analysis is unclear, both 2way ANOVA and One way ANOVA are mentioned in the legend for the same data. Please clarify.
- Discussion: The statement “As a result of these effects, rats exhibited a four-fold increase in natriuresis (Table 1).” is unclear. The numbers don’t show a four-fold increase. This is still stated, although in your response a correction was mentioned. Please revise.
Author Response
Reply to Reviewer # 1
We are most grateful to the Reviewer for the insightful critique. We apologize for oversights and we adjusted text as suggested. Text changes are highlighted.
1. Introduction: Your answer is appreciated. However, a goal is not a specific, testable hypothesis. Or is your study hypothesis-generating? Please provide this information.
|
This has been changed. Highlighted text, lines 72-73. Ref. 16.
|
2. Methods, contractile studies: The description of the test for successful removal of the endothelium is still missing, please add.
|
I apologize. We did not do endothelium removal. This text was pasted from a previous paper in which we studied human umbilical arteries with were incubated for 24 hours. PMID: 35328757, 4.5.1. Methods. Isolated Umbilical Artery Contractile Studies. |
3. Methods, statistics: The explanation regarding the definition of biological replicates is appreciated. However, it seems not to be added to the statistics section. Please add.
|
Done, lines 240-242. “Number of observations i.e. means of the number of animals/tissues that were analyzed by a given method. Each analyzed group contained 8 rats, however, the tissues of 5 animals from each were used for histochemistry, Western blotting, and other experiments.” |
4. Results, Fig. 2: You did not answer to my question: “As GAPDH was used for normalization, please provide evidence that PNx and MBG treatment does not alter GAPDH expression.” Please provide.
|
First, we measured the total protein content in all samples. Then we added the equal amount of total protein in each well (per lane) for electrophoresis. The optical density of GAPDH bands was reasonably equal for all samples – Sham, PNx and PNx+a/MBG (three GAPDH bands on each blot). If the optical densities of the bands (in the same amount of total protein) are equal, it means that the expression is not affected by the treatment. Finally, in contrast to GAPDH, we see the considerable differences in expression of studied proteins. If we wanted to study the expression of GAPDH protein itself, we would use the same approach – determine the optical density of GAPDH bands and calculated the ratio of GAPDH optical density to that of another normalizing protein, for example, beta-actin. However, in such a case the question on equal expression of beta-actin could be raised. In addition, this was published (GAPDH with MBG antibody and MBG. Antibodies: Am J Hypertens, 2022; 35(9): 828-832, doi: 10.1093/ajh/hpac065. Fig. 1; MBG: Am J Hypertens. 2022 Sep 1;35(9):828-832. doi: 10.1093/ajh/hpac065. Fig. 2. |
5. Results, Fig. 2: Your explanation that the PCR signal was normalized to 18S is appreciated. Please provide information that the expression of 18S is not affected by your treatments.
|
Information about this question has been added to the section Materials and Methods (line 328-331): PCR results demonstrate the same number of cycles during amplification of the 18S rRNA product in the three studied groups, the Ct level for 18s rRNA in the studied groups did not differ, which, as we believe, indicates that the injections of antibodies to MBG does not affect |
6. Results, Fig. 2: You did not answer to my previous question “For collagen and pro-collagen, was the treated group different from sham?”
|
Indeed, it was. This has been added to Fig. 2. Highlighted.
|
7. Results, Fig. 2: Your answer to my question “For the statement “Administration of aMBG mAbs to rats almost completely reversed the effect of PNx on Fli1, pro-collagen, and collagen-1 proteins.” p-values for the comparisons made should be provided (also for collagen PCR).” is unclear. For this statement the effect of PNx should be calculated (delta (PNx - sham) and compared with the effect of treatment. Otherwise, your statement is without justification.
|
Thank, you. The text was changed. Lines 119-120.
|
8. Results, Fig. 4: The statistical analysis is unclear, both 2way ANOVA and One way ANOVA are mentioned in the legend for the same data. Please clarify. |
In the revised text we used multiple comparisons test for two-way ANOVA. Figure 4, highlighted. |
9. Discussion: The statement “As a result of these effects, rats exhibited a four-fold increase in natriuresis (Table 1).” is unclear. The numbers don’t show a four-fold increase. This is still stated, although in your response a correction was mentioned. Please revise.
|
Sorry. A three-fold. Line 220, highlighted. |
Reviewer 2 Report
Comments and Suggestions for Authors
Thanks for answering the comments
Author Response
We are most grateful to the Reviewer for the insightful critique.